# Hydroxy-3-Phenylcoumarins as Multitarget Compounds for Skin Aging Diseases: Synthesis, Molecular Docking and Tyrosinase, Elastase, Collagenase and Hyaluronidase Inhibition, and Sun Protection Factor

**DOI:** 10.3390/molecules27206914

**Published:** 2022-10-15

**Authors:** Francesca Pintus, Sonia Floris, Antonella Fais, Benedetta Era, Amit Kumar, Gianluca Gatto, Eugenio Uriarte, Maria João Matos

**Affiliations:** 1Department of Life and Environmental Sciences, University of Cagliari, 09042 Monserrato, Italy; 2Department of Electrical and Electronic Engineering, University of Cagliari, 09123 Cagliari, Italy; 3Departamento de Química Orgánica, Facultade de Farmacia, Universidade Santiago de Compostela, 15782 Santiago de Compostela, Spain; 4Instituto de Ciencias Químicas Aplicadas, Universidad Autónoma de Chile, Santiago 7500912, Chile

**Keywords:** hydroxy-3-phenylcoumarins, tyrosinase, elastase, collagenase, hyaluronidase, sun protection factor, molecular docking, skin aging

## Abstract

Skin aging is a progressive biological process of the human body, and it is not only time-dependent. Differently substituted 3-phenylcoumarins proved to efficiently inhibit tyrosinase. In the current work, new substitution patterns have been explored, and the biological studies were extended to other important enzymes involved in the processes of skin aging, as elastase, collagenase and hyaluronidase. From the studied series, five compounds presented inhibitory activity against tyrosinase, one compound against elastase, eight compounds against collagenase and two compounds against hyaluronidase, being five compounds dual inhibitors. The 3-(4′-Bromophenyl)-5,7-dihydroxycoumarin (**1**) and 3-(3′-bromophenyl)-5,7-dihydroxycoumarin (**2**) presented the best profiles against tyrosinase (IC_50_ = 1.05 µM and 7.03 µM) and collagenase (IC_50_ = 123.4 µM and 110.4 µM); the 3-(4′-bromophenyl)-6,7-dihydroxycoumarin (**4**) presented a good inhibition against tyrosinase and hyaluronidase; the 3-(3′-bromophenyl)-6,7-dihydroxycoumarin (**5**) showed an effective tyrosinase and elastase inhibition; and 6,7-dihydroxy-3-(3′-hydroxyphenyl)coumarin (**11**) presented a dual profile inhibition against collagenase and hyaluronidase. Furthermore, considering the overall activities tested, compounds **1** and **2** proved to be the most promising anti-aging compounds. These compounds also showed to have a photo-protective effect, without being cytotoxic to human skin keratinocyte cells. To predict the binding site with the target enzymes, computational studies were also carried out.

## 1. Introduction

The skin is a very important and the largest human organ [1]. It is a crucial barrier and defense of the human body [2]. Tyrosinase, elastase, collagenase and hyaluronidase are enzymes directly involved in skin aging. Skin aging is a progressive biological process in human bodies, and it is not only time-dependent [3]. Indeed, it is mainly due to intrinsic (chronologic) and extrinsic aging (photo-aging). This second mechanism is due to the ultraviolet (UV) radiations’ exposure. This aging process brings changes to different components of the integumentary system. The epidermis becomes thinner, although the cell layer number does not change. Melanocytes, which contain melanin pigment, decrease in number and increase in size.

Melanin is the main substance responsible for skin color. It is also responsible for protecting the skin by absorbing UV sunlight up to 75% and scavenging reactive oxygen species (ROS). Tyrosinase is a copper-containing monooxygenase, responsible for catalyzing melanin synthesis within melanocytes [4]. It is one of the skin’s key regulatory enzymes. Facial hyperpigmentation disorders, such as melasma, post-inflammatory hyperpigmentation and solar freckles, are frequent skin conditions that can have a huge impact on patients’ quality of life and are often difficult to treat [5]. They are also related to skin aging disorders.

Elastin is a protein with elastic properties present in connective tissues, including the skin. It has an important role in keeping the tissue structure after stretch or recoil. Several studies suggest that skin aging may be directly related to the degradation of elastin [6]. Overexposure to UV irradiation and overproduction of ROS trigger elastase expression, which causes the hydroxylation of the dermal elastin fiber network [7]. Elastase inhibitors may increase the linearity of dermal elastic fibers and skin elasticity, preventing wrinkles and sagging. 

The outermost part of the skin is made up of fibroblasts and proteins including elastin and collagen. Collagen is the most abundant protein structure in the layer of human dermis, providing the skin’s traction resistance. This protein is one of those responsible for the plumpness, integrity and flexibility of the skin, keeping it youthful and healthy. Collagenase is the enzyme responsible for degrading collagen, and ROS are involved in activating this dermal enzyme [8]. Collagenase inhibitors may also prevent premature skin aging.

Hyaluronic acid plays a significant role in the reduction in wrinkles, wound healing and maintaining smooth and hydrated skin. The enzyme hyaluronidase catalyzes the hydrolysis of hyaluronic acid, decreasing the viscosity of body fluids and increasing the permeability of connective tissues. Hyaluronidase inhibitors are, therefore, efficient regulating agents, maintaining the equilibrium between the anabolism and catabolism of hyaluronic acid, keeping skin moist as well as smooth [9].

Tyrosinase, elastase, collagenase and hyaluronidase inhibitors, acting together, may prevent skin aging and its signs, such as dyspigmentation, wrinkles, sallowness and freckles [9]. The cosmetic market related to skin aging is continuously seeing strong growth. Therefore, the research into anti-aging and anti-depigmentation agents is a trend area of research. In our research group, we previously described interesting tyrosinase inhibitors based on the chemical structures of coumarins and polyphenols (Figure 1) [10,11,12,13,14]. Tyrosine-like condensed derivatives have been described as tyrosinase inhibitors. In particular, 3-amino-7-hydroxycoumarin presented an IC_50_ of 50 µM, being one of the best simple coumarin derivatives we described so far (Figure 1) [12]. Exploring the different substitution patterns on the 3-phenylcoumarin scaffold, 3-(4′-bromophenyl)-5,7-dihydroxycoumarin became an interesting tyrosinase inhibitor, with an IC_50_ of 1.05 µM (Figure 1) [11]. This molecule has been the inspiration for the current work, exploring the activity of bromine and hydroxy-substituted 3-phenylcoumarins on other enzymes involved in skin aging and skin diseases.

## 2. Results and Discussion

### 2.1. Chemistry

Molecules **1**–**17** were prepared following the synthetic strategies represented in Figure 1. These compounds were obtained by a Perkin–Oglialoro reaction, which is one of the most interesting reactions to obtain 3-phenylcoumarins presenting different hydroxyl groups. This reaction occurs in two consecutive steps: synthesis of the acetoxy precursors, followed by acidic hydrolysis. Acetoxy-3-phenylcoumarins were initially synthesized following a condensation reaction. Performing its traditional protocol, ortho-hydroxybenzaldehydes and phenylacetic acids reacted for around 16 h, at reflux temperature, using as reactants potassium acetate (CH_3_CO_2_K) and acetic anhydride (Ac_2_O). Performing at mild conditions, acetylation of the hydroxyl groups and closure of the pyrone ring occur at the same time. Further hydrolysis of the acetoxy compounds has been carried out in the presence of aqueous hydrochloric acid (HCl) and methanol (MeOH), at reflux temperature, for 3 h. This versatile synthetic methodology allows for the obtaining of hydroxyl substituted 3-phenylcoumarins (**1**–**17**) in very good yields.

### 2.2. Biological Assays

#### 2.2.1. Enzyme Inhibition

The synthetic molecules have been studied against different targets: tyrosinase, elastase, collagenase and hyaluronidase. The inhibitory profiles have been obtained by calculating the percentage of inhibition (Appendix A) and the IC_50_ values (Table 1). 

From the studied series, five compounds presented activity against tyrosinase, one compound against elastase, eight compounds against collagenase and two compounds against hyaluronidase. Individually, the best compound against tyrosinase is 3-(4′-bromophenyl)-5,7-dihydroxycoumarin (**1**) with an IC_50_ of 1.05 µM, 17 times more active than the standard inhibitor, kojic acid (IC_50_ = 17.9 µM). The best compound against elastase is 3-(3′-bromophenyl)-6,7-dihydroxycoumarin (**5**) with an IC_50_ of 37.4 µM, in the same range as the standard inhibitor oleanolic acid (IC_50_ = 25.7 µM). This molecule resulted in being the only one displaying this activity. The best compound against collagenase is 3-(4′-hydroxyphenyl)benzo[f]coumarin (**13**) with an IC_50_ of 96.8 µM, 1.25 times more active than the reference compound, epigallocatechin gallate (IC_50_ = 120.8 µM). Finally, the best compound against hyaluronidase is 3-(4′-bromophenyl)-6,7-dihydroxycoumarin (**4**), with an IC_50_ of 112.0 µM, being 1.9 times more active than the reference compound, oleanolic acid (IC_50_ = 212.4 µM).

Analyzing the multitarget profile of the molecules, five compounds proved to be dual inhibitors. The 3-(4′-Bromophenyl)-5,7-dihydroxycoumarin (**1**) and 3-(3′-bromophenyl)-5,7-dihydroxycoumarin (**2**) presented the best profiles against tyrosinase (IC_50_ = 1.05 µM and 7.0 µM) and collagenase (IC_50_ = 123.4 µM and 110.4 µM). The 3-(4′-Bromophenyl)-6,7-dihydroxycoumarin (**4**) presented a good inhibition against tyrosinase (IC_50_ = 39.9 µM) and hyaluronidase (IC_50_ = 112.0 µM), and 3-(3′-bromophenyl)-6,7-dihydroxycoumarin (**5**) showed an effective tyrosinase (IC_50_ = 8.25 µM) and elastase (IC_50_ = 37.38 µM) inhibition. Finally, 6,7-dihydroxy-3-(3′-hydroxyphenyl)coumarin (**11**) is a dual collagenase (IC_50_ = 143.3 µM) and hyaluronidase inhibitor (IC_50_ = 227.7 µM). 

#### 2.2.2. Structure–Activity Relationship (SAR)

Regarding the tyrosinase activity, the presence of a bromine in meta or para positions of the 3-phenyl ring seems important, together with at least a hydroxyl group at position seven of the coumarin scaffold (compounds **1**, **2**, **4**, **5** and **10**). The 3-(3′-Bromophenyl)-7-hydroxycoumarin (**10**), with a single hydroxyl at position seven of the coumarin scaffold, is the less active of the five active compounds. The other four molecules (compounds **1**, **2**, **4** and **5**) present two hydroxyl groups on their structures, in contiguous carbons (catechol at 6/7 positions) or non-contiguous carbons (at 5/7 positions), this last combination being the most interesting for this activity. 

The best compound against collagenase is 3-(4′-hydroxyphenyl)benzo[f]coumarin (**13**), being a naphthalin derivative. However, all the other naphthalin derivatives (compounds **14**–**17**) were inactive. Analyzing the substituents on the coumarin scaffold, three of the top five compounds, presenting activities very similar to the reference compound, present two hydroxyl groups at positions five and seven (compounds **1**–**3**). Analyzing the eight active compounds against this enzyme, five of these molecules present a bromine at ortho, meta or para positions of the 3-phenyl ring (compounds **1**–**3**, **6** and **8**). The other three (compounds **11**–**13**) present a hydroxyl group, each one in a different position. Therefore, no clear correlation between the nature and the position of the substitution patterns may be extrapolated. 

Finally, regarding hyaluronidase, the two active compounds present catechol groups at 5/6 positions of the coumarin scaffold (compounds **4** and **5**), bearing a bromine in para or ortho positions of the 3-phenyl ring, respectively.

#### 2.2.3. Sun Protection Factor (SPF)

In addition to the enzyme inhibition, the sun protection factor (SPF) has been calculated for all the molecules to investigate an additional photo-protective effect of the compounds (Table 2). Since UV rays are responsible for triggering oxidative stress reactions and progressive skin aging, it could be important to determine the SPF value of compounds with possible skin application. In fact, SPF indicates the capacity of a compound to absorb UV radiation and therefore protect skin from UV toxic effects. The SPF values found for the studied hydroxy-3-phenylcoumarins are between 4.27 and 8.23 and are reported in Table 2.

Eight compounds from the series have a value higher than 6, three of these dual enzymatic inhibitors (compounds **1**, **4** and **5**). The 3-(3′-Bromophenyl)-6,7-dihydroxycoumarin (**5**) presented the best dual profile against tyrosinase (IC_50_ = 8.3 µM) and elastase (IC_50_ = 37.4 µM), as well as the best SPF (8.23). The 3-(4′-Bromophenyl)-6,7-dihydroxycoumarin (**4**) presented the best dual profile against tyrosinase (IC_50_ = 39.9 µM) and hyaluronidase (IC_50_ = 112.0 µM), as well as the best SPF (8.23). Structurally, both compounds **4** and **5** are catechol derivatives, as well as 3-(2′-bromophenyl)-6,7-dihydroxycoumarin (**6**), with the next SPF value (8.08). The position of the bromine at the 3-phenyl ring seems irrelevant for this activity, as compounds **4**, **5** and **6** are ortho, meta and para derivatives, respectively. However, if this group is substituted for a hydroxyl (compounds **11** and **12**), the SPF is slightly lower (SPF = 5.44 and 5.64, respectively).

#### 2.2.4. Cell Viability

Considering the overall activities against all of the four aging-related enzymes, compounds **1** and **2** resulted in being the best molecules. In fact, among the compounds having dual activities, they possessed the best effect against tyrosinase and collagenase, but were also active against elastase and hyaluronidase. Therefore, compounds **1** and **2** showed the overall greatest effect on all the four tested enzymes (Table 1 and Appendix A). Thus, we focused our attention on these promising compounds with anti-aging properties. As a result, their effect on cell viability was evaluated, and molecular docking studies were also performed.

The results concerning the biosafety of compounds **1** and **2** are reported in Figure 2. The results obtained support the fact that both of the compounds are non-cytotoxic on the human skin keratinocyte cell line HaCaT up to the highest concentrations tested for enzymatic activities.

### 2.3. Molecular Docking

Molecular docking is a computational method widely applied to predict binding modes of small molecules (ligand) or macromolecules in contact with a receptor (protein). From docking experiments, the ligand’s affinity to the target protein can be predicted by analyzing the conformation and orientation of the ligand in the binding site. In this work, docking simulations have been performed for the most promising compounds **1** and **2** against their best target enzymes (collagenase and tyrosinase).

Regarding the binding with collagenase, docking analysis of compounds **1** and **2** were compared with compound **13**, the latter being the best collagenase inhibitor among the series (Figure 3).

The docking results showed that compound **13** displayed the best affinity with an energy score of −6.4 kcal/mol, followed by compound **2** and compound **1** (Table 3). The predicted energy trend from molecular docking is consistent with the trend noted in IC_50_ values obtained from the experiments.

In Figure 3, the interaction picture of compounds **1**, **2** and **13** with the binding site residues of collagenase obtained using the PLIP web tool is depicted [15]. The interaction with binding site residues L181, V215 and Y240 are conserved for compound **1** and compound **2**. Interactions with active-site residues N180 and R214 are found only for compound **1**, while interactions with residues G179 and H218 are found for compound **2**. On the contrary, compound **13**, with the best score, displays a different trend in amino acid distributions at the binding site, involving interactions with residues A182, A184 and H222. In a recent study [16], authors have reported residues A182, A184 and H222 to be important for ligand binding to the enzyme.

The docking results for the tyrosinase enzyme showed that compound **1** displayed the best affinity with an energy score of −5.4 kcal/mol, followed by compound **2** and compound **5** (Table 4). Here again, the predicted trend in energy scores from molecular docking is consistent with the trend noted in IC_50_ values obtained from the experiments. In Figure 4, the interaction picture of compounds **1**, **2** and **5** is illustrated, with the binding site residues of tyrosinase. 

It is evident from Figure 4 that compound **1** displays a richer interaction network with the binding site residues compared to compounds **2** and **5**. However, from the protein-ligand interaction picture, we can observe that some residues (E256, H263, F264 and V283) are conserved in the three investigated compounds. Conversely, H-phobic interaction with H85 is common to compound **1** and compound **2**, while H-bond interactions with V283 and N260 are present only with compound **1**. The importance of N260 for ligand binding to the enzyme has been reported previously [17]. These differences could explain a better docking score for compound **1**, which also displays the best IC_50_ value.

## 3. Materials and Methods

### 3.1. Chemistry

#### 3.1.1. General Information

All of the reagents used in the synthesis of the compounds were purchased from Sigma-Aldrich and used without further purification. All the solvents were of a commercially available grade. All the reactions were performed under argon atmosphere, unless otherwise mentioned. The reaction mixtures were purified by flash column chromatography using Silica Gel high purity grade (Merck grade 9385 pore size 60 Å, 230–400 mesh particle size), and analyzed by analytical thin-layer chromatography (TLC) using plates precoated with silica gel (Merck 60 F254, 0.25 mm). Their visualization was accomplished with UV light (254 nm) or potassium permanganate (KMnO_4_). ^1^H NMR and ^13^C NMR spectra were recorded on a Bruker AMX spectrometer at 250 and 75.47 MHz in the stated solvents (CDCl_3_ or DMSO-*_d6_*) using tetramethylsilane (TMS) as an internal standard. Chemical shifts were reported in parts per million (ppm) on the δ scale from an internal standard (NMR descriptions: s, singlet; d, doublet; dd, double doublet; t, triplet; m, multiplet). Mass spectroscopy was carried out using a Hewlett-Packard 5988 A spectrometer.

#### 3.1.2. General Procedure for the Synthesis of Acetoxy-3-Phenylcoumarins

Acetoxy-3-phenylcoumarins were synthesized under anhydrous conditions, using material previously dried at 60 °C for at least 12 h and at 300 °C for few minutes immediately before use. A solution containing anhydrous CH_3_CO_2_K (2.94 mmol), phenylacetic acid (1.67 mmol), and the corresponding hydroxysalicylaldehyde (1.67 mmol), in Ac_2_O (1.2 mL), was refluxed for 16 h. The reaction mixture was cooled, neutralized with 10% aqueous NaHCO_3_ and extracted with EtOAc (3 × 30 mL). The organic layers were combined, washed with distilled water, dried (anhydrous Na_2_SO_4_) and evaporated under reduced pressure. The product was purified by recrystallization in EtOH and dried to afford the desired compound.

#### 3.1.3. General Procedure for the Synthesis of Hydroxy-3-Phenylcoumarins (**1**–**17**)

Compounds **1**–**17** were obtained by hydrolysis of their acetoxylated counterparts, respectively. The appropriate acetoxylated coumarin, mixed with 2 N aqueous HCl and MeOH, was refluxed for 3 h. The resulting reaction mixture was cooled in an ice-bath and the reaction product, obtained as solid, was filtered, washed with cold distilled water, and dried under vacuum to afford the desired compound.

3-(4′-Bromophenyl)-5,7-dihydroxycoumarin (**1**) [11].3-(3′-Bromophenyl)-5,7-dihydroxycoumarin (**2**). Yield: 89%. Rf (Hexane/Ethyl Acetate 5:5): 0.17. Mp: 273–275 °C. ^1^H NMR (DMSO-*_d6_*) δ (ppm), *J* (Hz): 6.20–6.27 (m, 2H, H-6, H-8), 7.35 (t, 1H, H-5′, *J* = 7.6), 7.52 (d, 1H, H-6′, *J* = 7.6), 7.65 (d, 1H, H-4′, *J* = 7.6), 7.86 (s, 1H, H-2′), 8.08 (s, 1H, H-4), 10.48 (s, 1H, OH), 10.78 (s, 1H, OH). ^13^C NMR (DMSO-*_d6_*) δ (ppm): 93.9, 98.5, 102.4, 121.5, 125.5, 127.2, 130.5, 130.8, 133.3, 136.9, 137.9, 156.0, 156.6, 160.2, 162.7. MS *m/z* (%): 335.1 (16), 333.9 (90), 332.0 (92).3-(2′-Bromophenyl)-5,7-dihydroxycoumarin (**3**). Yield: 89%. Rf (Hexane/Ethyl Acetate 5:5): 0.15. Mp: 233–235 °C. ^1^H NMR (DMSO-*_d6_*) δ (ppm), *J* (Hz): 6.24–6.27 (m, 2H, H-6, H-8), 7.27–7.42 (m, 3H, H-4′, H-5′, H-6′), 7.66 (d, 1H, H-3′, *J* = 7.7), 7.81 (s, 1H, H-4), 10.57 (s, 1H, OH), 10.85 (s, 1H, OH). ^13^C NMR (DMSO-*_d6_*) δ (ppm): 94.0, 101.7, 110.3, 120.3, 123.7, 127.9, 130.2, 132.1, 132.5, 132.5, 136.8, 138.2, 156.3, 156.4, 162.5. MS *m/z* (%): 335.1 (17), 333.9 (96), 332.0 (91).3-(4′-Bromophenyl)-6,7-dihydroxycoumarin (**4**). Yield: 82%. Rf (Hexane/Ethyl Acetate 5:5): 0.06. Mp: 260–262 °C. ^1^H NMR (DMSO-*_d6_*) δ (ppm), *J* (Hz): 6.74 (s, 1H, H-8), 7.02 (s, 1H, H-5), 7.58–7.61 (m, 4H, H-2′, H-3′, H-5′, H-6′), 8.11 (s, 1H, H-4), 9.47 (s, 1H, OH), 10.29 (s, 1H, OH). ^13^C NMR (DMSO-*_d6_*) δ (ppm): 102.4, 111.6, 121.0, 121.3, 130.4, 131.2, 134.7, 137.7, 141.5, 143.3, 148.3, 150.9, 160.2. MS *m/z* (%): 335.1 (16), 334.0 (100), 333.0 (18), 332.0 (99).3-(3′-Bromophenyl)-6,7-dihydroxycoumarin (**5**). Yield: 86%. Rf (Hexane/Ethyl Acetate 5:5): 0.06. Mp: 255–257 °C. ^1^H NMR (DMSO-*_d6_*) δ (ppm), *J* (Hz): 6.73–6.76 (m, 1H, H-8), 7.02–7.05 (m, 1H, H-5), 7.31–7.49 (m, 1H, H-6′), 7.50 (d, 1H, H-5′, *J* = 8.5), 7.66 (d, 1H, H-4′, *J* = 8.5), 7.86–7.89 (m, 1H, H-2′), 8.16 (s, 1H, H-4), 9.50 (s, 1H, OH), 10.31 (s, 1H, OH). ^13^C NMR (DMSO-*_d6_*) δ (ppm): 102.3, 111.5, 121.5, 127.4, 130.4, 130.7, 130.8, 130.8, 137.8, 139.8, 142.1, 143.2, 148.4, 151.1, 166.0. MS *m/z* (%): 335.0 (16), 333.9 (99), 332.9 (14), 332.0 (100).3-(2′-Bromophenyl)-6,7-dihydroxycoumarin (**6**). Yield: 86 Rf (Hexane/Ethyl Acetate 5:5): 0.08. Mp: 248–250 °C. ^1^H NMR (DMSO-*_d6_*) δ (ppm), *J* (Hz): 6.67 (s, 1H, H-8), 7.33 (s, 1H, H-5), 7.38–7.42 (m, 3H, H-4′, H-5′, H-6′), 7.65–7.68 (m, 1H, H-3′), 7.86 (s, 1H, H-4), 9.46 (s, 1H, OH), 10.29 (s, 1H, OH). ^13^C NMR (DMSO-*_d6_*) δ (ppm): 102.6, 110.9, 112.6, 123.6, 123.7, 127.8, 130.2, 130.7, 132.0, 136.9, 143.2, 150.8, 153.9, 156.7, 164.6. MS *m/z* (%): 385.1 (12), 384.2 (30), 383.1 (100).3-(4′-Bromophenyl)-7,8-dihydroxycoumarin (**7**). Yield: 90%. Rf (Hexane/Ethyl Acetate 5:5): 0.14. Mp: 230–233 °C. ^1^H NMR (DMSO-*_d6_*) δ (ppm), *J* (Hz): 6.78–6.83 (m, 1H, H-6), 7.00–7.09 (m, 1H, H-5), 7.55–7.71 (m, 4H, H-2′, H-3′, H-5′, H-6′), 8.12 (s, 1H, H-4), 9.38 (s, 1H, OH), 10.15 (s, 1H, OH). ^13^C NMR (DMSO-*_d6_*) δ (ppm): 113.0, 119.5, 127.8, 128.3, 130.5, 131.2, 132.2, 136.1, 139.7, 143.4, 147.9, 150.0, 157.5. MS *m/z* (%): 335.1 (16), 334.0 (100), 332.9 (18), 332.0 (98).3-(3′-Bromophenyl)-7,8-dihydroxycoumarin (**8**). Yield: 86%. Rf (Hexane/Ethyl Acetate 5:5): 0.11. Mp: 252–254 °C. ^1^H NMR (DMSO-*_d6_*) δ (ppm), *J* (Hz): 6.82 (d, 1H, H-6, *J* = 8.3), 7.09 (d, 1H, H-5, *J* = 8.3), 7.37 (t, 1H, H-6′, *J* = 7.7), 7.55 (d, 1H, H-5′, *J* = 7.7), 7.70 (d, 1H, H-4′, *J* = 7.7), 7.90 (s, 1H, H-2′), 8.19 (s, 1H, H-4), 9.46 (s, 1H, OH), 10.23 (s, 1H, OH). ^13^C NMR (DMSO-*_d6_*) δ (ppm): 108.0, 112.8, 113.1, 119.6, 120.3, 121.6, 127.5, 130.4, 130.9, 132.0, 137.7, 147.6, 150.1, 150.8, 166.5, 169.3. MS *m/z* (%): 335.1 (16), 334.0 (100), 333.0 (18), 332.0 (98).3-(2′-Bromophenyl)-7,8-dihydroxycoumarin (**9**) [18].3-(3′-Bromophenyl)-7-hydroxycoumarin (**10**). Yield: 72%. Rf (Hexane/Ethyl Acetate 5.5): 0.09. Mp: 224–225 °C. ^1^H NMR (DMSO-*_d6_*) δ (ppm), *J* (Hz): 6.71–6.79 (m, 2H, H-6, H-8), 7.34–7.67 (m, 4H, H-5, H-4′, H-5′, H-6′), 7.86 (s, 1H, H-2′), 8.19 (s, 1H, H-4′), 10.67 (s, 1H, OH). ^13^C NMR (DMSO-*_d6_*) δ (ppm): 101.9, 112.0, 113.7, 120.6, 121.6, 127.4, 130.4, 130.4, 130.9, 137.6, 142.1, 153.3, 155.2, 160.0, 161.7. MS *m/z* (%): 318.0 (99), 317.1 (21), 316.0 (100).6,7-Dihydroxy-3-(3′-hydroxyphenyl)coumarin (**11**) [19].6,7-Dihydroxy-3-(2′-hydroxyphenyl)coumarin (**12**). Yield: 80%. Rf (Hexane/Ethyl Acetate 5:5): 0.05. Mp: 265–267 °C. ^1^H NMR (DMSO-*_d6_*) δ (ppm), *J* (Hz): 6.76–6.88 (m, 3H, H-5, H-8, H-5′), 6.97–7.03 (m, 1H, H-4′), 7.18–7.22 (m, 2H, H-2′, H-3′), 7.82 (s, 1H, H-4), 9.40 (s, 1H, OH), 9.46 (s, 1H, OH). ^13^C NMR (DMSO-*_d6_*) δ (ppm): 102.4, 111.4, 112.4, 115.8, 118.8, 123.0, 129.3, 131.1, 142.5, 143.0, 148.2, 150.1, 155.2, 155.6, 166.7. MS *m/z* (%): 392.1 (16), 391.1 (29), 390.0 (100).3-(4′-Hydroxyphenyl)benzo[*f*]coumarin (**13**) [20].3-(3′-Hydroxyphenyl)benzo[*f*]coumarin (**14**) [20].3-(2′-Hydroxyphenyl)benzo[*f*]coumarin (**15**) [20].3-(3′,4′-Dihydroxyphenyl)benzo[*f*]coumarin (**16**) [20]. 3-(3′,5′-Dihydroxyphenyl)benzo[*f*]coumarin (**17**) [20]. 

### 3.2. Biological Assays

#### 3.2.1. Enzyme Inhibition

Results from all the inhibition tests described below were reported as a percentage of the blank control. Since the compounds were dissolved in DMSO, in the assays carried out without compounds, DMSO was added to the reaction mixture as blank control. The IC_50_ values, concentrations of compounds resulting in 50% inhibition of enzymatic activity, were determined by interpolating the dose–response curves. Data from activity experiments were recorded using an Ultrospec 2100 spectrophotometer (Biochrom Ltd., Cambridge, UK).

##### Tyrosinase Inhibition Assay

The inhibition of tyrosinase was monitored by using DOPA as substrate. The reaction solution contained phosphate buffer (25 mM, pH 6.8) and mushroom tyrosinase (100 U/mL), with or without test samples. Then, after the addition of L-DOPA (0.5 mM) into the mixture, the activity was monitored at 475 nm, absorbance of the dopachrome product. Kojic acid was used as a positive control.

##### Elastase Inhibition Assay

The inhibition of elastase was determined by measuring the production of *p*-nitroaniline during the reaction of the enzyme with the *N*-succ-(Ala)3-nitroanilide (SANA) substrate. The assay was carried out in Tris-HCl buffer (0.1 M, pH 8.0) containing porcine pancreatic elastase (3.3 µg/mL). The reaction mixture was incubated in the presence or in the absence of the test compounds for 20 minutes and, once the substrate (1.6 mM) was added, the enzyme activity was monitored at 410 nm. Oleanolic acid was used as a positive control.

##### Collagenase Inhibition Assay

Collagenase from *Clostridium histolyticum* (1 U/mL) was prepared in 0.05 M Tricine buffer (pH 7.5), with 0.4 M NaCl and 0.01 M CaCl_2_, and incubated with different concentrations of test samples for 15 minutes. After adding the *N*-(3-[2-Furyl]-acryloyl)-Leu-Gly-Pro-Ala (FALGPA) substrate, prepared in the same buffer solution reaching a final concentration of 0.8 mM, the absorbance was registered at 340 nm. Epigallocatechin gallate was used as a positive control.

##### Hyaluronidase Inhibition Assay

Inhibition of hyaluronidase was determined following the method reported by Chompoo and co-authors [21]. A 5 μL test sample was incubated with 100 μL of enzyme solution (1.5 U) containing 20 mM sodium phosphate buffer (pH 7.0) with 77 mM NaCl and bovine serum albumin (BSA) 0.01%. After 10 min at 37 °C, 100 μL of the substrate solution (0.03% hyaluronic acid in 300 mM sodium phosphate, pH 5.35) was added and the reaction mixture was incubated further for 45 minutes at 37 °C. Hyaluronic acid (undigested) was precipitated with 1 mL of acid albumin solution, containing BSA 0.1 % in sodium acetate (24 mM) and acetic acid (79 mM, pH 3.75). After 10 minutes at room temperature, the absorbance was monitored at 600 nm. Oleanolic acid was used as a positive control.

##### In Vitro Determination of the Sun Protection Factor

The sun protection factor of coumarins’ derivatives was measured by the UV absorbance method, as previously reported [22]. The absorbances of the compounds (0.1 mg/mL) were registered in the range of 290–320 nm, with 5 nm increments, and three measurements were performed at each point. The SPF was determined by using the Mansur Equation [23]:$$\mathrm{SPF}=\mathrm{CF}\;\times \;\Sigma_{290}^{320}\;\times \;\mathrm{EE}(\lambda )\;\times \;I(\lambda )\;\times \;\mathrm{Abs}(\lambda )$$
where CF = correction factor (10); EE (λ) = erythemogenic effect of radiation with wavelength λ; I(λ) = solar intensity spectrum; Abs(λ) = spectrophotometric absorbance values at wavelength λ. The values of EE (λ) × I(λ) are constant as determined by Sayre et al. (Table 5) [24].

##### Cell Culture

Human skin keratinocyte cell line (HaCaT; CLS–Cell Lines Service, Eppelheim, Germany) was cultured in Dulbecco’s Modified Eagle’s Medium (DMEM) with 10% fetal bovine serum (FBS; Gibco, NY, USA) and 1% penicillin/streptomycin at 37 °C in a humidified atmosphere with 5% CO_2_. Cell viability was determined by the MTT (3-(4,5-dimethylthiazol-2-yl)-2,5-diphenyltetrazolium bromide) colorimetric assay, as previously reported [22]. Briefly, HaCaT cells were seeded in 96-well plates and incubated with compounds at different concentrations (0–200 µM). After 24 h, cells were labelled with MTT solution for 3 h at 37 °C. DMSO was added to the formed violet formazan precipitates and the absorbance of each well was measured at 560 nm.

### 3.3. Docking Simulations

The molecular docking method was applied to investigate the interaction between compounds **1**, **2** and **13** and collagenase enzyme, and between compounds **1**, **2** and **5** and tyrosinase enzyme. The three-dimensional (3D) structures of enzymes were obtained from protein data bank (PDB): collagenase (PDB id: 1CGL) [25], and tyrosinase (PDB id: 2Y9X) [26]. A Marvin JS tool [27] was used to sketch the two-dimensional structure and then converted into 3D using OpenBabel [28] software tools. The obtained 3D structures of the compounds were geometry optimized [29] within the framework of density functional theory, a popular quantum mechanical modeling method. The optimized structures were saved in Mol2 format to be then used for the docking experiment. Docking was carried out using COACH-D webserver [30] (last accessed on 3 May 2022). Concisely, the COACH algorithm applies five different methods to provide a consensus prediction of protein-binding site residues, and then the ligands from the user are docked into the predicted binding pockets. In the final step, the ligand-binding conformations are refined with AutoDock Vina [31], a very effective molecular docking algorithm.

## 4. Conclusions

The present study revealed the inhibitory activities of some coumarin derivatives against aging-related enzymes. All of the compounds were screened against tyrosinase, elastase, collagenase and hyaluronidase, and some of them were active toward one or more target enzymes. In order to identify multitarget molecules, 3-(4′-bromophenyl)-5,7-dihydroxycoumarin (**1**) and 3-(3′-bromophenyl)-5,7-dihydroxycoumarin (**2**) represented the most promising compounds, being active against all the four enzymes, with the best inhibitory effects against tyrosinase and collagenase. These compounds presented the best profiles against tyrosinase being 17 and 2.6 times, respectively, more active than the standard inhibitor, kojic acid. In addition, compounds **1** and **2** showed a good inhibitory effect against collagenase with IC_50_ values in the same range as the standard inhibitor, epigallocatechin gallate. Molecular docking studies allowed for the identification of the most probable ligand-binding site and the important amino acid residues for the interactions. Moreover, these compounds showed no toxicity on human skin keratinocyte cells and an interesting photoprotective effect. Since UV radiation triggers oxidative stress and aging-related enzymes activation, this effect results in a further indirect antioxidant activity and enzymatic inhibition. All these properties may be enhanced, for example, with nanoparticle formulation administration systems to improve bioavailability and applicability to the skin. Therefore, all the results pointed out compounds **1** and **2** as promising molecules with anti-aging properties for potential cosmeceutical drug development.

## Data Availability

Not applicable.

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
