# Peer review of "Hydroxy-3-Phenylcoumarins as Multitarget Compounds for Skin Aging Diseases: Synthesis, Molecular Docking and Tyrosinase, Elastase, Collagenase and Hyaluronidase Inhibition, and Sun Protection Factor"

_molecules, 2022, doi:10.3390/molecules27206914_

Round 1

Reviewer 1 Report

Dear authors,

Your manuscript is well written and clear.

In conclusion, I should have appreciated more perspectives concerning the identified molecules and their use for an anti-aging claim.

Best regards. 

Author Response

Author’s response to the Reviewer

We thank the Reviewer for his comments and suggestions. We have addressed carefully all the points raised.

We have highlighted all these changes in the manuscript, as requested. For convenience, the responses to Reviewer’s Remarks are shown in blue

Point-by-point response:

Your manuscript is well written and clear.

We are thankful to the Reviewer for appreciating our work.

In conclusion, I should have appreciated more perspectives concerning the identified molecules and their use for an anti-aging claim.

We thank the Reviewer for the suggestion. We have included the future perspective in the manuscript.

Reviewer 2 Report

The authors have done an Interesting work. The authors have synthesized coumarin derivatives and evaluated against skin aging related enzymes. I suggest the authors to include Structure-Activity Relationship under a separate heading. Also, conclusion should reflect the compounds with best activity as compared to standards for each enzyme.

Author Response

Author’s response to the Reviewer

We thank the Reviewer for his comments and suggestions. We have addressed carefully all the points raised.

We have highlighted all these changes in the manuscript, as requested. For convenience, the responses to Reviewer’s Remarks are shown in blue.

Point-by-point response:

The authors have done an Interesting work.

The authors are grateful to the Reviewer for finding our work to be interesting.

The authors have synthesized coumarin derivatives and evaluated against skin aging related enzymes.

I suggest the authors to include Structure-Activity Relationship under a separate heading.

As suggested, we included, as a separate heading, the Structure-Activity Relationship.

Also, conclusion should reflect the compounds with best activity as compared to standards for each enzyme.

We have modified the conclusions accordingly.
